# Hybrid Projective Synchronization of Fractional-Order Extended Hindmarsh–Rose Neurons with Hidden Attractors

**Xuerong Shi**  **and Zuolei Wang** *

School of Mathematics and Statistics, Yancheng Teachers University, Yancheng 224002, China
* Correspondence: wzlyctc@163.com

**Abstract:** In view of the diversity of stimulated current that neurons may experience, an extended Hindmarsh–Rose neuron model is proposed and the corresponding fractional-order neuron model, with no equilibrium point, is depicted. Additionally, various hidden attractors of the addressed neuron model are analyzed by changing system parameters and the order of fractional-order neuron system. Furthermore, hybrid projective synchronizations of the proposed neurons are investigated and schemes are obtained by designing suitable controllers according to fractional stability theory. Besides, the validity of the theoretical results is verified through numerical simulations. In short, the research results have potential application in revealing the dynamical behaviors of neuron system and controlling the behaviors of neuron into certain status.

**Keywords:** hybrid projective synchronization; fractional-order; extended Hindmarsh–Rose neuron; hidden attractor; fractional stability theory

**MSC:** 34H05



## 1. Introduction

The biological nervous system, which is essential within an organism, is responsible for some important functions. It can not only sense external stimuli, but also generate, process, conduct, and integrate signals and engage with various cognitive activities, such as feeling, learning, thinking, and controlling the movement of the organism. It is well known that the nervous system is composed of billions of neurons. Research on the complex dynamics of neuron plays an important role in revealing the dynamics of nervous system and can provide scientific principles for defeating various kinds of neurological diseases.

Mathematical models of neurons have great value in exactly describing the electrical activity of neurons. For this reason, neuron models have been built from different perspectives. In 1952, Alan Lloyd Hodgkin and Andrew Fielding Huxley established the Hodgkin–Huxley neuron model (HH) [1] based on the electrical activity of squid giant axons, which pioneered computational neuroscience and provided an important starting point for theoretical biological neuroscience and computational neuroscience. Since the mid-20th century, models of neurons' electrical activity have been successively deduced, featuring the Hindmarsh–Rose (HR) [2], FitzHugh–Nagumo [3], Chay [4], Morris–Lecar model [5], and Izhikevich [6] models. Meanwhile, their dynamics have also been investigated [7–11]. These neuron models are convenient for simplified calculation and detailed nonlinear analysis. However, in the process of simplification, some factors in the neuron model are ignored or simplified. Recently, Wang and Zhang established a novel neuron model from the perspective of energy [12], which had similar dynamics to HH model [13]. The phase synchronization of function neurons developed from a simple neuron circuit was discussed and the energy diversity of neurons exposed to different situations was revealed [14]. With the establishment and development of neuron models, neurodynamics was applied into many fields, such as the variant model of FitzHugh–Nagumo model and its applications in pulse-stream neural networks [15], dynamics of the modified Izhikevich

neuron model and the associated electronic realization applications [16]. Especially, in the connection with some diseases, neurodynamics played an important role [17–21]. For example, to reveal the mechanism of Parkinson's disease, an extended thalamic–basal ganglia model was proposed [17] and a biophysically based model was utilized to demonstrate the parkinsonian dynamical properties [18]; a hippocampal neuron-reduced model under the pathological condition of Alzheimer's disease was built for AD research [19]; the neurodegenerative disorder Huntington's disease (HD) was discussed via observing focal adhesion kinase (FAK) activity [20]; and a disease model of amyotrophic lateral sclerosis (ALS) was produced by analyzing the histopathological changes and the number of motor neurons [21].

Among many neuron models, the HR neuron model has attracted strong interest from many researchers due to its unique advantages. For example, the effect of synapses on the electrical activity and synchronization status of Hindmarsh–Rose neurons under Gaussian white noise was revealed [17]. Besides, an adaptive Hindmarsh–Rose neuron model was addressed and system parameters were estimated via a learning algorithm for synchronization between neurons [18]. The existence of Hopf bifurcation of time-delay fractional-order Hindmarsh–Rose neuron was obtained via a linearizing method [22]. Additionally, dynamical behaviors of delayed Hindmarsh–Rose neurons near nonhyperbolic equilibrium were discussed. It was found that time delay changing could induce saddle-node bifurcation [23]. The multistability of a modified neuron model was considered and spatiotemporal patterns of the neuron network composed of the considered neuron were explored [24]. The effect of external current on the synchronization of a disordered Hindmarsh–Rose neural network was revealed. It was found that whether the network can obtain synchronization or desynchronization was dependent on the type of noise [25]. Topology and dynamical pattern in neuron network with HR neurons as the nodes can be locally and accurately recognized by using a deterministic learning algorithm and constructing a fast dynamical pattern recognition method via synchronization, respectively [26]. Given the environment the neuron is in, by introducing a new variable denoting magnetic flux, the improved Hindmarsh–Rose neuron models were proposed, their dynamics have been investigated, and some results have been gained. Changes in magnetic flux can alter the number of equilibrium points and their stability [27]. A neuron network composed of modified neuron coupled with discontinuous exponential flux was constructed and factors affecting the spatiotemporal pattern were reported [28]. The effect of electromagnetic induction could influence the neuron electric activities [29]. Results in [30] suggest that the dynamics of the neuron model is greatly affected by phase noise. From the perspective of energy, a pair of delayed Hindmarsh–Rose neurons coupling with electrical and chemical synapses was considered, causing the total energy consumption to be weirdly different [31]. According to the above, various dynamical behaviors and different kinds of synchronization about integer-order neuronal systems have been investigated.

With the extensive application of fractional calculus, fractional-order neuron models have been proposed and their dynamics have been explored. For example, various dynamical behaviors of a fractional-order memristive HR neuron induced by small parameter changing were discussed [32] and Hopf bifurcation of fractional-order HR neuron model caused by time delay was investigated [33]. Different kinds of synchronizations of fractional-order neurons were discussed under different situations, such as adaptive synchronization under electromagnetic radiation [34], generalized projective synchronization [35], synchronization of fractional-order neurons in presence of noise [36], and synchronization of nonidentical fractional-order neurons [37]. By analyzing the related literature, it can be found that, although some dynamics of fractional-order neurons have been investigated, there is some work to be performed due to the complex behaviors of neurons closely related to the nervous function of an organism. For example, as an important synchronization means, chaos projective synchronization was hardly reported in the existing results. As a vital phenomenon, hidden attractors of neurons should be investigated.

Therefore, in this paper, to further research the dynamics of neurons under a complicated external forcing current, a fractional-order extended HR neuron model is put forward and its dynamics are explored. The proposed neuron model can be more accurately fit the neuronal system to a certain extent because of the introduction of fractional calculus. The research into the addressed neuron model is helpful for revealing the complex phenomenon of the nervous system and provides some theoretical basis for the application of neurodynamics in biological information. The remainder of this paper is arranged as follows. Fractional stability theory and relative preliminaries are introduced in Section 2, hidden attractors of the mentioned neuron model are analyzed in Section 3, and in Section 4, schemes about hybrid projective synchronizations are given. The validity of the schemes is verified via theoretical analysis and numerical simulation. Conclusions are drawn in Section 5.

## 2. Fractional Stability Theory and Relative Preliminaries

Compared with other fractional calculus, Riemann–Liouville, Grünwald–Letnikov, Atangana–Baleanu, and so on, the Caputo derivative can avoid problems with uncertainties and singularities while modeling fractional-order systems. Due to the super-singularity, the Riemann–Liouville fractional derivative is not convenient for engineering and physical modeling. To solve this problem, the Italian geophysicist Caputo proposed a fractional differential definition with weak singularity. It solved the fractional-order initial value problem in the Riemann–Liouville definition and was widely used in the modeling of viscoelastic materials. Therefore, in following discussion, the Caputo derivative will be used, with the definition is depicted as:

**Definition 1.** *For a given continuously differentiable function $f(t) : [0, +\infty) \to R$, Caputo's fractional differential operator [38] can be depicted as*

$$
{}^{c}_{0}D^{q}_{t}f(t) = \frac{1}{\Gamma(n-q)} \int_{0}^{t} \frac{f^{n}(\tau)}{(t-\tau)^{q-n+1}} d\tau \tag{1}
$$

*where $0 < q < 1$, $\Gamma(\cdot)$ is the gamma function $\Gamma(\tau) = \int_{0}^{\infty} t^{\tau-1} e^{-t} dt$ with the characteristics of $\Gamma(\tau+1) = \tau\Gamma(\tau)$.*

*Since the definition of the Caputo fractional differential operator was proposed, it has been widely used due to its similar Laplace transformation formula to that of the integer-order operator. For brevity, ${}^{c}_{0}D^{q}_{t}f(t)$ is abbreviated as $D^{q}f(t)$.*

**Lemma 1** ([39]). *Take the system*

$$
D^{q}X = A(X)X \tag{2}
$$

*into consideration, where $X = (x_1, x_2, \cdots, x_n) \in R^n$ is the system variable, $0 < q < 1$. If all eigenvalues $\lambda_i(i = 1, 2, \cdots, n)$ of $A(X)$ at the equilibrant point $X*$ meet the condition $|\arg(\lambda_i)| \geq \frac{q\pi}{2}$, then System (2) will stabilize at the equilibrant point $X*$.*

**Note 1.** *For the equilibrium point $X*$ of System (2), if all eigenvalues of $A(X*)$ have a negative real part, it can be said that System (2) is stable at $X*$.*

**Lemma 2** ([39]). *Take the system*

$$
D^{q}X = A(X)X \tag{3}
$$

*into account with the state variable $X \in R^n$ and order $q$ in the range of $0 < q < 1$; if there is a positive definite matrix $P$, $X^T P D^q X \leq 0$ holds for any $X$. It can be said that system (3) will stabilize at zero.*

**Lemma 3.** *If $A$ is a real symmetric matrix, then there exists a sufficiently large real numbert, such that $tI + A$ is a positive definite matrix, where $I$ is the identity matrix with the same type as $A$.*

**Proof.** Supposing $A = \begin{pmatrix} a_{11} & a_{12} & \cdots & a_{1n} \\ a_{21} & a_{22} & \cdots & a_{2n} \\ \vdots & \vdots & \ddots & \vdots \\ a_{n1} & a_{n2} & \cdots & a_{nn} \end{pmatrix}$ with $a_{ij} = a_{ji}$, then we have

$$tI + A = \begin{pmatrix} k + a_{11} & a_{12} & \cdots & a_{1n} \\ a_{21} & k + a_{22} & \cdots & a_{2n} \\ \vdots & \vdots & \ddots & \vdots \\ a_{n1} & a_{n2} & \cdots & k + a_{nn} \end{pmatrix} \tag{4}$$

which has a leading principal submatrix with $k$-order of

$$\Delta_k(t) = \begin{vmatrix} t + a_{11} & a_{12} & \cdots & a_{1k} \\ a_{21} & t + a_{22} & \cdots & a_{2k} \\ \vdots & \vdots & \ddots & \vdots \\ a_{k1} & a_{k2} & \cdots & t + a_{kk} \end{vmatrix} \tag{5}$$

When $t$ is large enough, $\Delta_k(t)$ is the determinant of a strictly principal diagonally dominant matrix, namely $t + a_{ii} > \sum_{j \neq i} |a_{ij}| (i = 1, 2, \cdots, n)$ holds. Therefore, one can obtain $\Delta_k(t) > 0 (k = 1, 2, \cdots, n)$. This suggests that $tI + A$ is positive definite. □

## 3. System Description and the Hidden Attractors

Recalling the famous HR neuron model [40], it is described as

$$\begin{cases} \dot{x} = y - ax^3 + bx^2 - z + I_{ext} \\ \dot{y} = c - dx^2 - y \\ \dot{z} = r(S(x + x_0) - z) \end{cases} \tag{6}$$

where $x$, $y$, and $z$ are system variables representing the membrane potential, recovery variable, and adaption current, respectively. Other symbols, including $a$, $b$, $c$, $d$, $r$, $S$, $x_0$, $I_{ext}$ are system parameters with $x_0$ representing resting potential, $I_{ext}$ an external forcing current, $a$, $b$ describing the rate of membrane potential change, $c$, $d$ being utilized to keep the adaption current in voltage-clamp conditions, and $r$ and $S$ relating to a short depolarizing current characterizing the change in adaption current.

For the external forcing current $I_{ext}$, dynamics of the HR neuron model with direct current have been widely discussed [41–44]. In neuroscience, some noninvasive transcranial stimulation can generate various currents affecting the electricity activity of neurons. Meanwhile, with the gradual deepening of research on neuron models, it is found that the classical HR neuron model is subject to some restrictions in exploring complex characteristic of neurons. Based on this, an extended HR neuron model is attained by substituting $I_{ext} = mx\tanh(\varphi)$ in (6) and is written as

$$\begin{cases} \dot{x} = y - ax^3 + bx^2 - z + mx\tanh(\varphi) \\ \dot{y} = c - dx^2 - y \\ \dot{z} = r(S(x + x_0) - z) \\ \dot{\varphi} = -kx \end{cases} \tag{7}$$

where variable $\varphi$ enters to express exchange of calcium ions.

According to Definition 1, the corresponding fractional-order extended HR neuron model in terms of the Caputo definition can be given as

$$
\begin{cases}
D^q x = y - ax^3 + bx^2 - z + mx\tanh(\varphi) \\
D^q y = c - dx^2 - y \\
D^q z = r(S(x + x_0) - z) \\
D^q \varphi = -kx
\end{cases}
\tag{8}
$$

Let $y - ax^3 + bx^2 - z + mx\tanh(\varphi) = 0$, $c - dx^2 - y = 0$, $r(S(x + x_0) - z) = 0$, $-kx = 0$, it is known that, when $k \neq 0$, if and only if $Sx_0 = c$, the equilibrium point of System (8) can be obtained as $(0, c, c, \varphi_0)$ with $\varphi_0$ being any real number. In this situation, there are infinitely many equilibrium points for System (8). The stability of the equilibrium point $(0, c, c, \varphi_0)$ is closely dependent on the eigenvalues of

$$
J = \begin{pmatrix}
-3ax^2 + 2bx + m\tanh(\varphi) & 1 & -1 & mx\sec h^2(\varphi) \\
-2dx & -1 & 0 & 0 \\
rS & 0 & -r & 0 \\
-k & 0 & 0 & 0
\end{pmatrix}
$$

Which is the corresponding Jacobian matrix of System (8). While the eigenvalues are closely related to the value of $\varphi_0$, therefore, for this condition, the stability of the equilibrium point $(0, c, c, \varphi_0)$ is very complex and will be discussed in another work.

Obviously, for non-zero values of $a$, $b$, $c$, $d$, $r$, $S$, $x_0$, $m$, and $k$, with $Sx_0 \neq c$, System (8) has no equilibrium point. In the following discussion, we choose $a = 1.0$, $b = 3.0$, $c = 1.0$, $d = 5.0$, $r = 0.006$, $S = 4.0$, $x_0 = 0.6$, $m = 0.9$, $k = 0.1$, fractional-order $q$ changing from 0.6 to 0.378, dynamics of System (8) are calculated via the Bashforth–Moulton predictor-corrector algorithm and given in Figures 1–4, from which it can be determined that System (8) appears to have various hidden attractors with different values of order $q$. It suggests that System (8) is provided with hidden dynamics of period-doubling bifurcation leading to chaos with $q$ decreasing from 0.6 to 0.378.

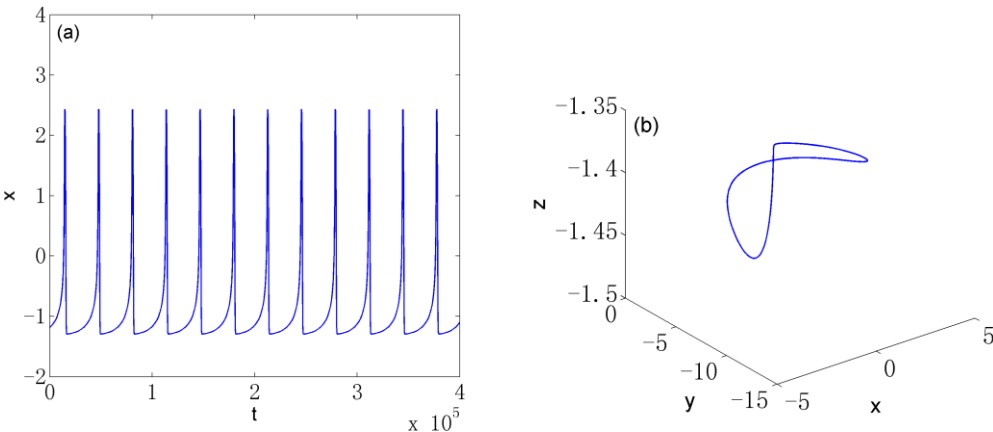

**Figure 1.** Hidden period-1 attractor of System (8) when $q = 0.6$: (**a**) time evolution of variable $x$; (**b**) phase portrait in 3D space.

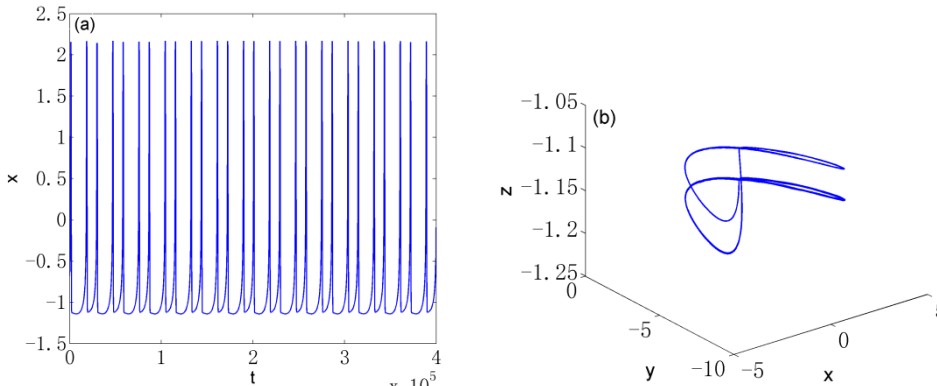

**Figure 2.** Hidden period-2 attractor of system (8) when $q = 0.39$: (**a**) time evolution of variable $x$; (**b**) phase portrait in 3D space.

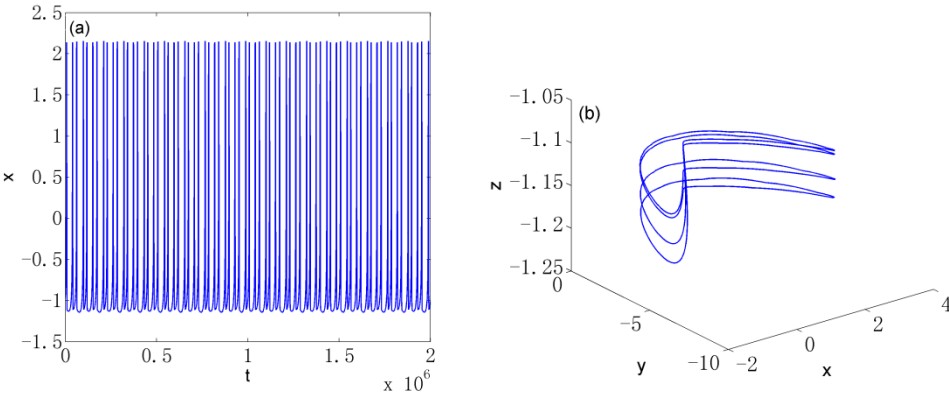

**Figure 3.** Hidden period-4 attractor of System (8) when $q = 0.3828$: (**a**) time evolution of variable $x$; (**b**) phase portrait in 3D space.

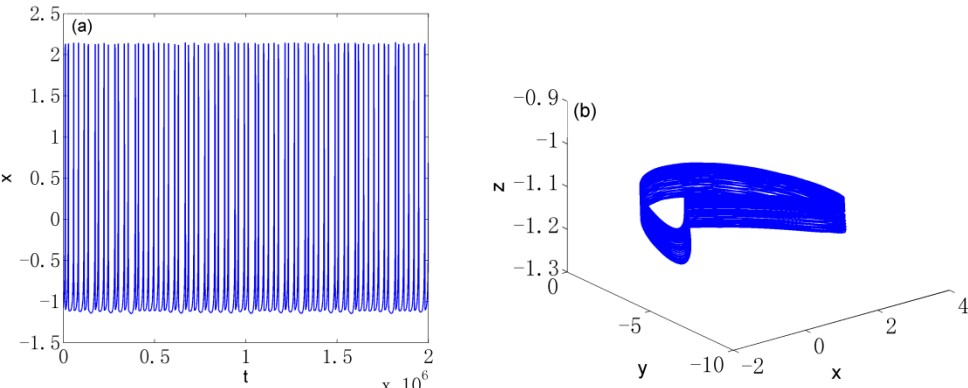

**Figure 4.** Hidden chaotic attractor of System (8) when $q = 0.378$: (**a**) time evolution of variable $x$; (**b**) phase portrait in 3D space.

Furthermore, we select $a = 1.0$, $b = 3.0$, $c = 1.0$, $d = 5.0$, $r = 0.006$, $S = 4.0$, $x_0 = 0.6$, $m = 0.2$, $q = 0.6$; with different values of $k$, dynamics of System (8) are computed for $k = 0.2$ and $k = 0.1$, which are drawn in Figure 5 (hidden periodic-1 attractor) and Figure 6 (hidden multi-periodic attractor), respectively. As shown in Figures 5 and 6, different hidden attractors can also be induced by the alteration of parameter $k$.

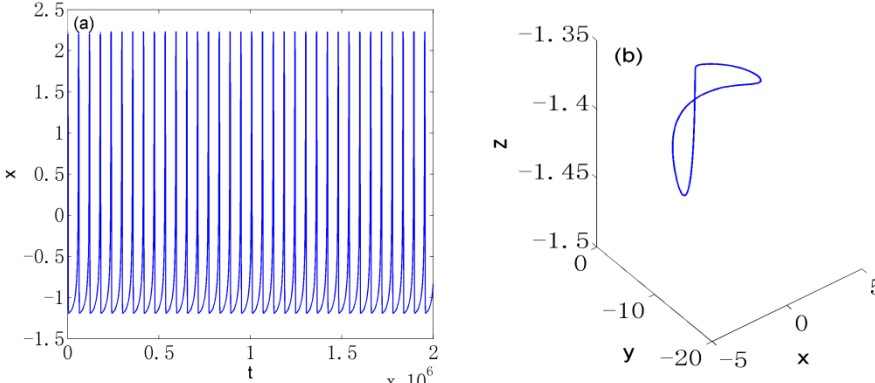

**Figure 5.** Hidden periodic-1 attractor of System (8) when $k = 0.2$: (**a**) time evolution of variable $x$; (**b**) phase portrait in 3D space.

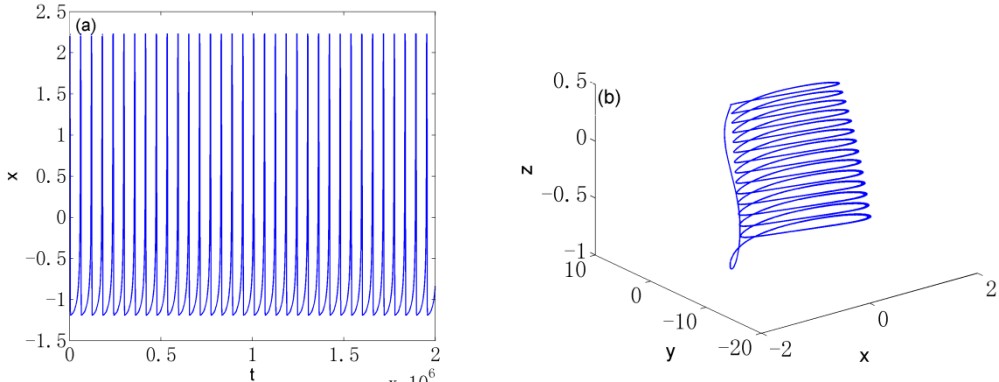

**Figure 6.** Hidden multi-periodic attractor of System (8) when $k = 0.1$: (**a**) time evolution of variable $x$; (**b**) phase portrait in 3D space.

From Figures 1–6 one conclusion can be drawn that System (8) appears to have complex hidden dynamics. By choosing suitable system parameters, certain dynamical phenomenon of neuron System (8) can be achieved.

## 4. Hybrid Projective Synchronization of Fractional-Order Extended HR Neurons

### 4.1. Hybrid Projective Synchronization Schemes

In this section, hybrid projective synchronization schemes are mainly investigated via designing effective controllers. To this end, the master system is taken as

$$\begin{cases} D^q x_1 = y_1 - a x_1^3 + b x_1^2 - z_1 + m x_1 \tanh(\varphi_1) \\ D^q y_1 = c - d x_1^2 - y_1 \\ D^q z_1 = r(S(x_1 + 0.6) - z_1) \\ D^q \varphi_1 = -k x_1 \end{cases} \tag{9}$$

and the corresponding slave system with controllers is given as

$$\begin{cases} D^q x_2 = y_2 - a x_2^3 + b x_2^2 - z_2 + m x_2 \tanh(\varphi_2) + u_1 \\ D^q y_2 = c - d x_2^2 - y_2 + u_2 \\ D^q z_2 = r(S(x_2 + 0.6) - z_2) + u_3 \\ D^q \varphi_2 = -k x_2 + u_4 \end{cases} \tag{10}$$

with controllers $u_1$, $u_2$, $u_3$, and $u_4$ to be determined.

By constructing the controllers, hybrid projective synchronizations can be realized and main results are given as the following theorems.

**Theorem 1.** *Define the error system between (9) and (10) as*

$$
\begin{cases}
e_1 = x_2 - \alpha_1 x_1 \\
e_2 = y_2 - \alpha_2 y_1 \\
e_3 = z_2 - \alpha_3 z_1 \\
e_4 = \varphi_2 - \alpha_4 \varphi_1
\end{cases}
\tag{11}
$$

*where $\alpha_i (i = 1, 2, 3, 4)$ are positive values, which are called projection factors. If controllers in response System (10) are chosen as*

$$
\begin{cases}
u_1 = -f_1 - k_1 e_1 \\
u_2 = -f_2 - k_2 e_2 \\
u_3 = -f_3 - k_3 e_3 \\
u_4 = -f_4 - k_4 e_4
\end{cases}
\tag{12}
$$

*where*

$$
f_1 = \alpha_2 y_1 + b x_2^2 - a x_2^3 - \alpha_3 z_1 + m x_2 \tanh(\varphi_2)
$$
$$
- \alpha_1 (y_1 + b x_1^2 - a x_1^3 - z_1 + m x_1 \tanh(\varphi_1))
$$

$$
f_2 = c - d x_2^2 - \alpha_2 (c - d x_1^2)
$$

$$
f_3 = r S \alpha_1 x_1 + 0.6 r S - r S \alpha_3 x_1 - 0.6 r S \alpha_3
$$

$$
f_4 = k(\alpha_4 - \alpha_1) x_1
$$

*Then, for suitable values of $k_1$, $k_2$, $k_3$, and $k_4$, error System (11) will be stable at zero. Essentially, a kind of hybrid projective synchronization can be realized.*

**Proof.** Finding the $q$ order derivative of (11) with respect to time $t$ can yield

$$
\begin{cases}
D^q e_1 = e_2 - e_3 + f_1 + u_1 \\
D^q e_2 = -e_2 + f_2 + u_2 \\
D^q e_3 = r S e_1 - r e_3 + f_3 + u_3 \\
D^q e_4 = -k e_1 + f_4 + u_4
\end{cases}
\tag{13}
$$

Substituting (12) into (13), we can gain

$$
\begin{cases}
D^q e_1 = e_2 - e_3 - k_1 e_1 \\
D^q e_2 = -e_2 - k_2 e_2 \\
D^q e_3 = r S e_1 - r e_3 - k_3 e_3 \\
D^q e_4 = -k e_1 - k_4 e_4
\end{cases}
\tag{14}
$$

Therefore, denoting $E = (e_1, e_2, e_3, e_4)^T$, one has

$$
E^T D^q E = e_1(e_2 - e_3 - k_1 e_1) + e_2(-e_2 - k_2 e_2) + e_3(r S e_1 - r e_3 - k_3 e_3) + e_4(-k e_1 - k_4 e_4)
$$
$$
= -k_1 e_1^2 + e_1 e_2 + (r S - 1) e_1 e_3 - k e_1 e_4 - (1 + k_2) e_2^2 - (r + k_3) e_3^2 - k_4 e_4^2
$$
$$
= -E^T P E
$$

where

$$
P =
\begin{pmatrix}
k_1 & -\frac{1}{2} & \frac{1-rS}{2} & \frac{k}{2} \\
-\frac{1}{2} & 1 + k_2 & 0 & 0 \\
\frac{1-rS}{2} & 0 & r + k_3 & 0 \\
\frac{k}{2} & 0 & 0 & k_4
\end{pmatrix}
\tag{15}
$$

According to Lemma 3, for large enough values of $k_1$, $k_2$, $k_3$, $k_4$, and $P$ is a positive definite matrix. This means that $E^T D^q E \leq 0$. In light of Lemma 2, error System (11) is stable at zero, which indicates that the above-mentioned hybrid projective synchronization of Systems (10) and (9) is achieved with projection factors $\alpha_i$ ($i = 1, 2, 3, 4$). Theorem 1 is proved. □

Likewise, the controllers (12) in Theorem 1 can be generalized to more general case and the corresponding results are depicted as following Theorems 2 and 3. Since the proof processes of them are similar to Theorem 1, the proof of Theorems 2 and 3 is omitted here and only the results are given.

**Theorem 2.** *Define the error system between Systems (10) and (9) as*

$$
\begin{cases}
e_1 = x_2 - \alpha_1 x_1 \\
e_2 = y_2 - \alpha_2 y_1 \\
e_3 = z_2 - \alpha_3 z_1 \\
e_4 = \varphi_2 - \alpha_4 (x_1 + y_1 + z_1)
\end{cases}
\tag{16}
$$

*if controllers in response System (10) are chosen as*

$$
\begin{cases}
u_1 = -f_1 - k_1 e_1 \\
u_2 = -f_2 - k_2 e_2 \\
u_3 = -f_3 - k_3 e_3 \\
u_4 = -f_5 - k_4 e_4
\end{cases}
\tag{17}
$$

*where*

$$
f_1 = \alpha_2 y_1 + b x_2^2 - a x_2^3 - \alpha_3 z_1 + m x_2 \tanh(\varphi_2) - \alpha_1 (y_1 + b x_1^2 - a x_1^3 - z_1 + m x_1 \tanh(\varphi_1))
$$

$$
f_2 = c - d x_2^2 - \alpha_2 (c - d x_1^2)
$$

$$
f_3 = r S \alpha_1 x_1 + 0.6 r S - r S \alpha_3 x_1 - 0.6 r S \alpha_3
$$

$$
f_5 = k \alpha_1 x_1 - \alpha_4 [-a x_1^3 + (b - d) x_1^2 + (r S - k) x_1 - (1 + r) z_1 + m x_1 \tanh(\varphi_1) + c + 0.6 r S]
$$

*Then, for appropriate values of $k_1, k_2, k_3, k_4$ error System (16) will stabilize to zero. That is, another kind of hybrid projective synchronization between the two systems can be achieved.*

**Theorem 3.** *Define*

$$
(e_1, e_2, e_3, e_4)^T = (x_2, y_2, z_2, \varphi_2)^T - (\Psi_1, \Psi_2, \Psi_3, \Psi_4)^T
\tag{18}
$$

*where*

$$
\begin{cases}
\Psi_1 = c_{11} x_1 + c_{12} y_1 + c_{13} z_1 + c_{14} \varphi_1 \\
\Psi_2 = c_{21} x_1 + c_{22} y_1 + c_{23} z_1 + c_{24} \varphi_1 \\
\Psi_3 = c_{31} x_1 + c_{32} y_1 + c_{33} z_1 + c_{34} \varphi_1 \\
\Psi_4 = c_{41} x_1 + c_{42} y_1 + c_{43} z_1 + c_{44} \varphi_1
\end{cases}
\tag{19}
$$

*if controllers in response System (10) are designed as*

$$
\begin{cases}
u_1 = -k_1 e_1 + c_{11} D^q x_1 + c_{12} D^q y_1 + c_{13} D^q z_1 + c_{14} D^q \varphi_1 - D^q x_2 \\
u_2 = -k_2 e_2 + c_{21} D^q x_1 + c_{22} D^q y_1 + c_{23} D^q z_1 + c_{24} D^q \varphi_1 - D^q y_2 \\
u_3 = -k_3 e_3 + c_{31} D^q x_1 + c_{32} D^q y_1 + c_{33} D^q z_1 + c_{34} D^q \varphi_1 - D^q z_2 \\
u_4 = -k_4 e_4 + c_{41} D^q x_1 + c_{42} D^q y_1 + c_{43} D^q z_1 + c_{44} D^q \varphi_1 - D^q \varphi_2
\end{cases}
\tag{20}
$$

*Then, for proper values of $k_1, k_2, k_3, k_4$, the solution of error System (18) will converge to zero. That is, a kind of more general hybrid projective synchronization can be obtained between Systems (9) and (10).*

*4.2. Numerical Simulation Verification*

To test the results in the above theorems, numerical simulations are conducted using the Adams–Bashforth–Moulton predictor–corrector algorithm in the MATLAB program. In the following simulations, system parameters are chosen as $a = 1.0$, $b = 3.0$, $c = 1.0$, $d = 5.0$, $r = 0.006$, $S = 4.0$, $m = 0.9$, $k = 0.1$. $k_1 = k_2 = k_3 = k_4 = 0.2$, $q = 0.378$. Initial values of the master system and slave system are selected as $(-2, -3, -6, 2)$ and

$(1, -2, 1, 0)$, respectively. The evolution of errors (11) with controllers (12) in Theorem 1 can be calculated with projection factors $\alpha_1 = 0.5$, $\alpha_2 = 1.5$, $\alpha_3 = 2$, $\alpha_4 = 5$ and are drawn in Figure 7. As indicated in Figure 7, the error System (11) will be stable at zero over time $t$. That is, the hybrid projective synchronization mentioned in Theorem 1 can be achieved. Figure 8 confirms this result. The curves of error System (16) with controllers (17) are described in Figure 9 with $\alpha_1 = 0.5$, $\alpha_2 = 1.5$, $\alpha_3 = 2$, $\alpha_4 = 1$, which means that error System (16) tends to zero as time changes. That is to say, the kind of hybrid projective synchronization mentioned in Theorem 2 can be realized. Figure 10 tests the result in Figure 9. For Theorem 3, projection factor matrix $C = (c_{ij})(i = 1, 2, 3, 4; j = 1, 2, 3, 4)$ is taken as

$$C = \begin{pmatrix} 1 & 2 & 3 & 1 \\ 2 & 1 & 2 & 3 \\ 2 & 1 & 1 & 1 \\ 3 & 1 & 1 & 1 \end{pmatrix} \tag{21}$$

Figure 11 depicts the time series of error System (18) with controllers (20) and Figure 12 presents the synchronization status of variables in Theorem 3.

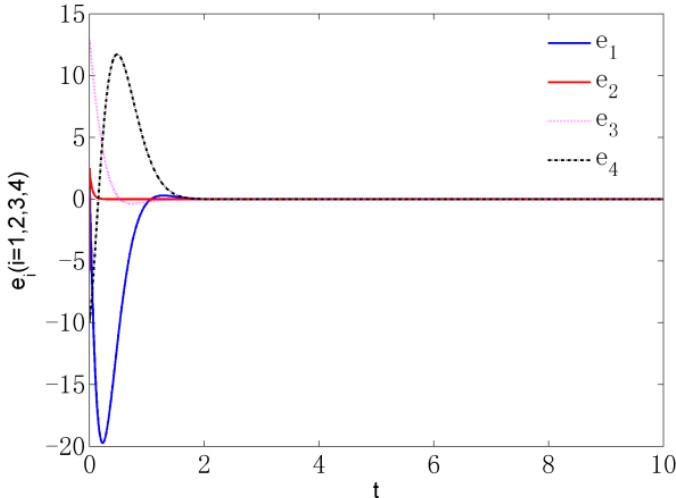

**Figure 7.** Evolution of error System (11) with projection factors $\alpha_1 = 0.5$, $\alpha_2 = 1.5$, $\alpha_3 = 2$, $\alpha_4 = 5$ in Theorem 1.

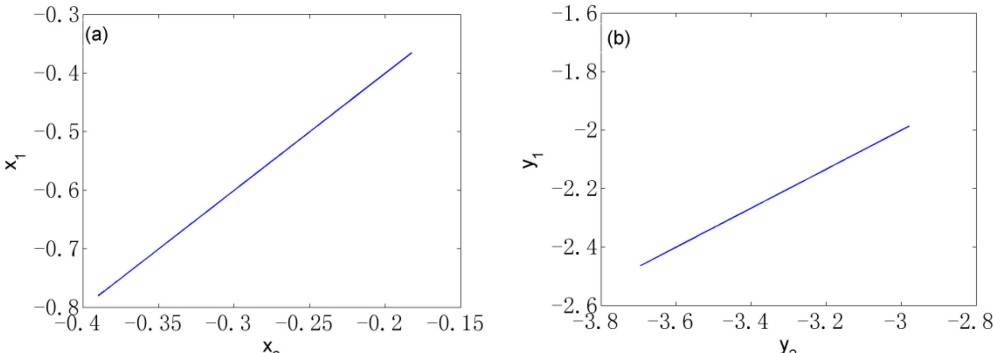

**Figure 8.** *Cont.*

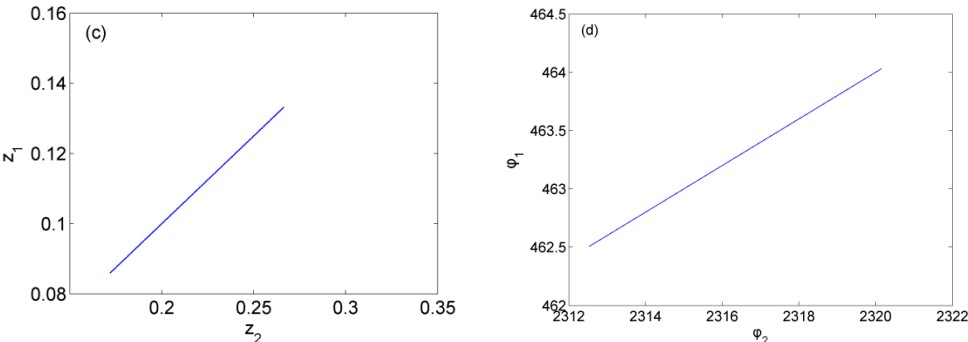

**Figure 8.** Synchronization status of variables with projection factors $\alpha_1 = 0.5$, $\alpha_2 = 1.5$, $\alpha_3 = 2$, $\alpha_4 = 5$ in Theorem 1. (**a**) Synchronization state between variables $x_1$ and $x_2$ with $\alpha_1 = 0.5$; (**b**) Synchronization state between variables $y_1$ and $y_2$ with $\alpha_2 = 1.5$; (**c**) Synchronization state between variables $z_1$ and $z_2$ with $\alpha_3 = 2$; (**d**) Synchronization state between variables $\varphi_1$ and $\varphi_2$ with $\alpha_4 = 5$.

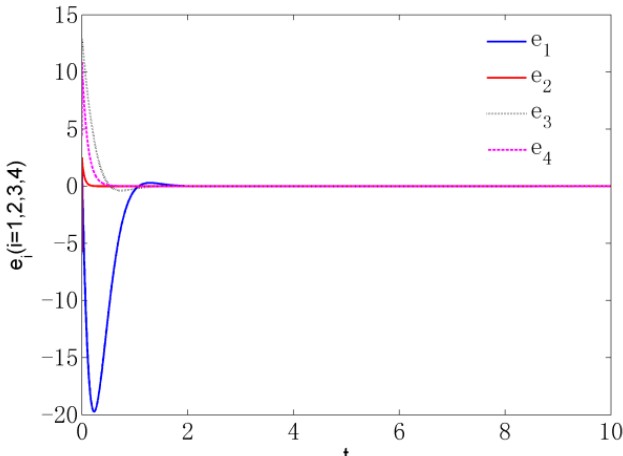

**Figure 9.** Evolution of error System (16) with projection factors $\alpha_1 = 0.5$, $\alpha_2 = 1.5$, $\alpha_3 = 2$, $\alpha_4 = 1$ in Theorem 2.

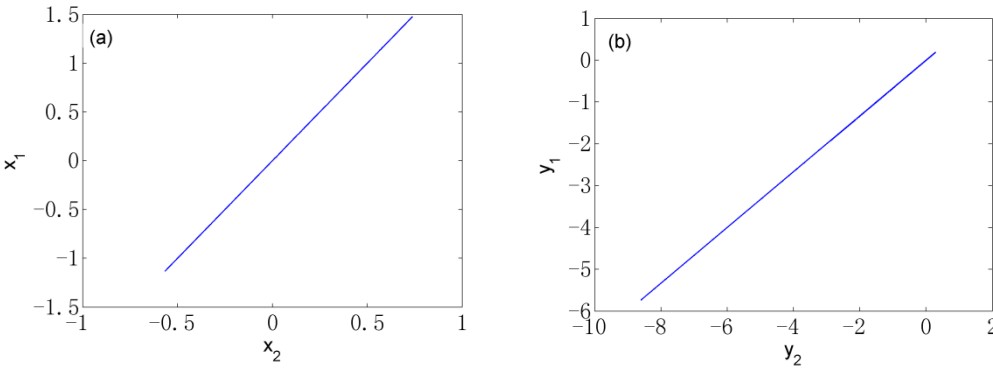

**Figure 10.** *Cont.*

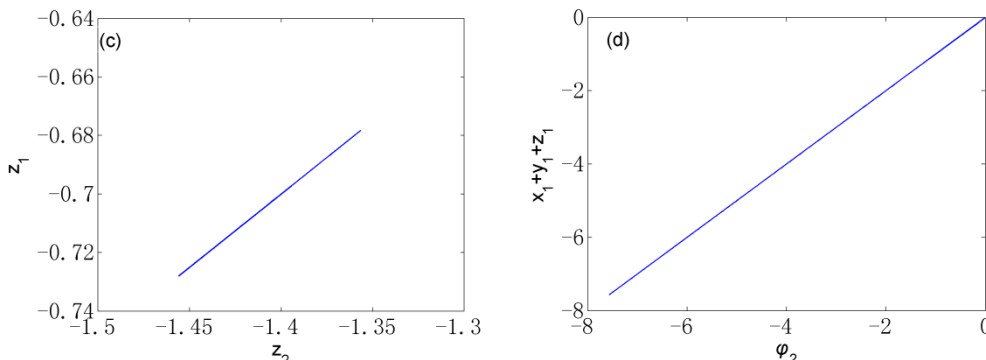

**Figure 10.** Synchronization status of variables with projection factors $\alpha_1 = 0.5$, $\alpha_2 = 1.5$, $\alpha_3 = 2$, $\alpha_4 = 1$ in Theorem 2. (**a**) Synchronization state between variables $x_1$ and $x_2$ with $\alpha_1 = 0.5$; (**b**) Synchronization state between variables $y_1$ and $y_2$ with $\alpha_2 = 1.5$; (**c**) Synchronization state between variables $z_1$ and $z_2$ with $\alpha_3 = 2$; (**d**) Synchronization state between variables $x_1 + y_1 + z_1$ and $\varphi_2$ with $\alpha_4 = 1$.

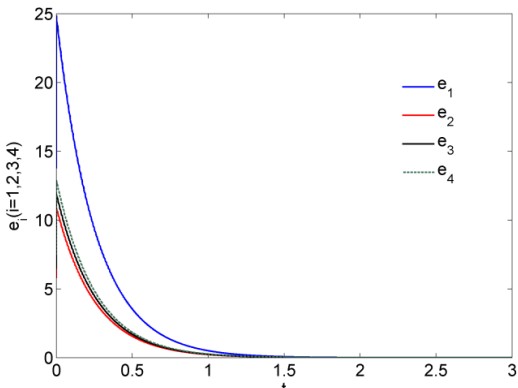

**Figure 11.** Evolution curve of error System (18) with projection factor matrix $C$ for Theorem 3.

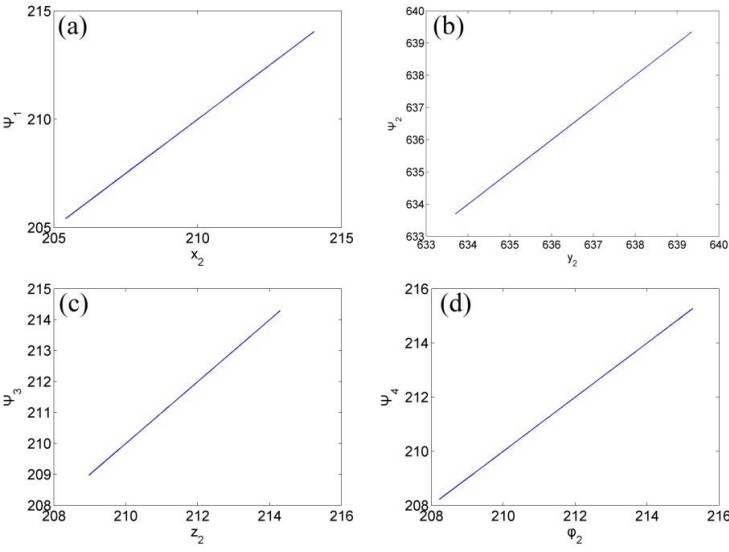

**Figure 12.** Synchronization status of variables with projection factor matrix (21) for Theorem 3. (**a**) Synchronization state between variables $\Psi_1$ and $x_2$ with projection coefficient $(1, 2, 3, 1)$; (**b**) Synchronization state between variables $\Psi_2$ and $y_2$ with projection coefficient $(2, 1, 2, 3)$; (**c**) Synchronization state between variables $\Psi_3$ and $z_2$ with projection coefficient $(2, 1, 1, 1)$; (**d**) Synchronization state between variables $\Psi_4$ and $\varphi_2$ with projection coefficient $(3, 1, 1, 1)$.

Figures 7–12 suggest that different kinds of hybrid projective synchronizations can be realized by designing suitable controllers.

## 5. Conclusions

In this work, based on the classical Hindmarsh—Rose neuron model with three variables, considering the variety of external stimulation current induced by noninvasive or minimally invasive surgery, a fractional-order extended HR neuron model is constructed. Numerical simulations indicates that the proposed model appears to show firing patterns similar to HR neurons. This means that the addressed model can be regarded as a kind of neuron model, namely the fractional-order extended HR neuron model. Then, some dynamical behaviors of the fractional-order extended HR neuron model are investigated and the main results can be depicted as follows.

(1)  The fractional-order extended Hindmarsh–Rose neuron has various hidden attractors with the change in system parameter or the order of fractional-order neuron models, such as period-1, period-2, period-4, chaotic, and multi-periodic attractors. Especially, the dynamics appear to have a phenomenon of period-doubling bifurcation leading to chaos with the decrease in order $q$. Compared with the traditional self-excited attractor, research into hidden attractors of neuron systems is of great significance for understanding the complexity of dynamical behavior of neuron systems and revealing the mechanisms of neurological disorder.

(2)  Three kinds of hybrid projective synchronization schemes are given by designing suitable controllers. In addition, the efficiency and feasibleness of the proposed schemes are verified via theoretical analysis and numerical simulation. According to the results, the addressed synchronization method is suitable for both simple projection factors and more complex projection factors. Compared with many kinds of chaos synchronization, projective synchronization is one of the most noticeable types of synchronization. This is because different state variables of projective synchronization synchronize to a scaling factor. This scaling feature can be used to extend binary numbers to m-decimal numbers for faster transmission in secure communications. Hybrid projective synchronization in our work can further improve the security of secure communications because of the adjustability of scaling factors and synchronization variables.

(3)  By utilizing a proper hybrid projective synchronization scheme and designing a projection factor, system variables can synchronize various variables or a combination of several different variables. That is to say, the dynamics of fractional-order extended Hindmarsh–Rose neurons can be controlled to the given status effectively. This result has potential applications in terms of the functional integration of neurons and is helpful for exploring the integration mechanism of neurons. For example, different properties of objects can be unified and presented as a whole after being processed in specific visual areas of the brain. This means that various neurons' function can be integrated by realizing the hybrid projective synchronization.

After decades of development, neuroscience is moving toward applications and has been used in various fields. Neural computing science will promote the development of related disciplines, especially in the emerging high-tech field, such as nanotechnology, bioinformatics, life sciences, information encryption. Meanwhile, the development of neuroscience is helpful for the diagnosis and treatment of neurological disorders.

As we all know, the nervous system consists of a large number and diversity of neurons with different biological functions. However, most neuron models consider only a single biological function and the external stimulus is often taken as a simple equivalent current. In the actual nervous system, many neurons may have various functions and are also stimulated by a variety of stimulus. Stimuli of light, sound, or mechanical stress can be considered when designing intelligent neuron sensors and processors, which gives new insights and guidance for exploring neurodynamics.

**Author Contributions:** Conceptualization, X.S. and Z.W.; methodology, X.S.; software, Z.W.; validation, X.S. and Z.W.; formal analysis, Z.W.; investigation, X.S.; resources, X.S.; writing—original draft preparation, X.S.; writing—review and editing, Z.W. All authors have read and agreed to the published version of the manuscript.

**Funding:** This work is funded by National Natural Science Foundation of China (Grant No 11872327) and Natural Science Research Project of Jiangsu Colleges and Universities (20KJA190001).

**Data Availability Statement:** Data is contained within the article.

**Conflicts of Interest:** The authors declare no conflict of interest.

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
