# Peer review of "Hybrid Projective Synchronization of Fractional-Order Extended Hindmarsh–Rose Neurons with Hidden Attractors"

_axioms, doi:10.3390/axioms12020157_

Round 1

Reviewer 1 Report

The article deals with the topical problem of modeling neuron systems using fractional-order analysis. The authors successfully apply Caputo derivatives and the fractional stability theory during their research. However, despite the significance of the considered problem, the following suggestions should be considered to clarify some issues:

1. The presented literature review should be presented more critically. Mainly, research gaps in previous studies were not presented.

2. It's well-known that the Caputo derivative avoids problems with uncertainties and singularities while modeling fractional-order systems. Nevertheless, despite other types of derivatives (e.g., Riemann-Liouville, Grünwald-Letnikov, Atangana-Baleanu, and so on), its introduction should be clarified more transparently.

3. The presented case study considers an extended Hindmarsh-Rose neuron model. However, ways for further applicability of such a model should be added in terms of its advantages (e.g., compared with models by Hodgkin-Huxley, Poisson-Nernst-Planck, and so on) and the practical significance of its further implementation.

4. Numerical simulations were carried out using the Adams-Moulton algorithm. However, compared with other algorithms, an analysis of the results' reliability was not provided. Moreover, the A-stability of the applied linear multistep method was not checked according to the 2nd Dahlquist barrier statement.

5. The obtained results need to be analyzed in terms of their consistency with other recent studies in modeling neuron systems. Therefore, a critical discussion is recommended before the conclusions.

6. The 3rd conclusion is declarative. The obtained results should supplement it.

Overall, the article can be recommended for publication after considering all the aforementioned suggestions.

Reviewer 2 Report

The authors proposed an extended Hindmarsh-Rose neuron model and depicted the corresponding fractional-order neuron model with no equilibrium point. They analysed various hidden attractors of the addressed neuron model by changing system parameters and order of fractional-order neuron system and the theoretical results has been validated and verified through numerical simulations.

Overall the manuscript is not well written. Many of the results and conclusions of this paper are quite basic. I strongly recommend to expand: Introduction, Results and the Conclusions/Discussion sections. The aim should be to:

(1) give a broader view of the literature on the topic and explain the current state-of-the-art of the topic where the paper is framed;

(2) clarify and discuss the motivation of this study, and the novelty and the significance of the results obtained here;

(3) compare them with those available in the literature, also including discussions on potential applications;

(4) complete the manuscript with some additional, less basic results;

(5) expand the Discussion to highlight the relevance and interest of this work for its aimed scientific community.

(6) Several mathematical errors found in the manuscript such as in equation (6) line 114.

(7) The authors have not studied the stability of the solution not appropriately.

(8) The authors have not checked the solution uniqueness and also not validated the obtained results.

 I cannot support publication unless the authors undertake all the above actions in full.

Reviewer 3 Report

Authors study and propose  a new approach of modelling of moving neurons. They specially study hybrid projective synchronization of fractional / order extended Hindmarsch / rose neurons with hidden attractors. They describe with the mentioned model dynamical behaviors of neuron system and controlling the behaviors of neuron into certain status. According my opinion the topic is chosen well, it is very useful and also used methods are chosen well.   Therefore I recommend the publication of the paper under minor revisions:

page/line

1.Introduction: The authors give the overview of the current situation of the models and research. Please can you include also some wider area of application and possibility of applications in medicine and biology? In the connection with some other diseases. 

3/99  Please delete a space behind "P" and a comma.

3/110 Please insert a space behind an ending bracelet   ")holds"  ") holds".

3/112-114. Please correct life sides of the equality (6)...there is some mess in symbols.

3/ last paragraph...116, 117, 118....Please correct formatting of the mathematical symbols.

3/117, 118  Please there are sentences beginning with symbols: "c, d" and "r and S". Please reformulate the sentence to not start with a mathematical symbol. Please correct in the whole text.

6/154, 156, 157...Please correct formatting of mathematical symbols...

6/157 Please insert  a space behind "Figure6(Hidden".

7/ 173..Please correct formatting of mathematical symbols.

7/182-186 and 8/212-216  Please correct fomatting of mathematical equations. 

8/245 Please insert a dot behind mathematical formula (21).

12/Conclusions....Please insert some visions and line into the future and also open problems.

Round 2

Reviewer 2 Report

The revised version of the manuscript can be accepted for publication.